# Automatic Emotion-Based Music Classification for Supporting Intelligent IoT Applications

**Yeong-Seok Seo** [1] and **Jun-Ho Huh** [2,*]

1   Department of Computer Engineering, Yeungnam University, Gyeongsan 38541, Korea; ysseo@yu.ac.kr
2   Department of Software, Catholic University of Pusan, Busan 46252, Korea
*   Correspondence: 72networks@pukyong.ac.kr or 72networks@cup.ac.kr; Tel.: +82-51-510-0662

**Abstract:** With the arrival of the fourth industrial revolution, new technologies that integrate emotional intelligence into existing IoT applications are being studied. Of these technologies, emotional analysis research for providing various music services has received increasing attention in recent years. In this paper, we propose an emotion-based automatic music classification method to classify music with high accuracy according to the emotional range of people. In particular, when the new (unlearned) songs are added to a music-related IoT application, it is necessary to build mechanisms to classify them automatically based on the emotion of humans. This point is one of the practical issues for developing the applications. A survey for collecting emotional data is conducted based on the emotional model. In addition, music features are derived by discussing with the working group in a small and medium-sized enterprise. Emotion classification is carried out using multiple regression analysis and support vector machine. The experimental results show that the proposed method identifies most of induced emotions felt by music listeners and accordingly classifies music successfully. In addition, comparative analysis is performed with different classification algorithms, such as random forest, deep neural network and K-nearest neighbor, as well as support vector machine.

**Keywords:** artificial intelligence; emotion; emotional intelligence; music classification; music analysis; human behavior; context-aware aspects; software engineering

## 1. Introduction

Internet of Things (IoT), based on various Information and Communication Technology (ICT), has emerged as a leader of the fourth industrial revolution, creating new products and services that never before existed. Because it is recognized as a core technology that maximizes convenience in the daily lives of humans and it suggests an optimal solution for different lifestyles and environments, there is significant interest in the changes that IoT will bring to the global society [1].

As the demand for cutting-edge IoT technology increases and extensive research is carried out, many researchers from industry, academia and research institutions are focusing on the development of emotional intelligence that is able to identify human thoughts and emotions. Emotional analysis is a research topic that is motivated by the innate desire to understand the laws of human emotions but is hindered by the difficulty to interpret the biometric data. In order to implement emotional expressions, even Facebook, Google, Microsoft and Apple, which are leading companies in the IT industry, are developing technologies to extract or learn emotions or are taking over companies engaged in this type of research [2,3].

One of the specific research areas for such emotional analysis is music. As the accessibility of a massive amount of multimedia content has increased, it has become easy to access and listen to music in our daily life. However, the genre or type of music that is desired can significantly vary

according to individual emotions or moods. Thus, technologies that analyze human emotions and then recommend music according to a certain atmosphere are being developed to promote ease of use. As A.I. speakers equipped with Amazon Alexa [4,5], Apple Siri [6–8], Google Assistant [9,10] and Microsoft Cortana [11,12] have recently been released and commercialized, it is possible to select music according to user moods [13]. In this way, music-related services such as the recommendation of suitable music will progressively expand by analyzing the emotions of people. In addition, it can be used by businesses to target customers in various industries such as marketing, education, entertainment, computer gaming and healthcare [14].

Therefore, in this paper, we propose an emotion-based automatic music classification method to classify music with high accuracy according to the emotional range of humans. In particular, when the new (unlearned) songs are added to a music-related IoT application, it is necessary to build mechanisms to classify them automatically according to the emotion of humans. This is one of the practical issues for the small and medium-sized enterprises that develop the applications. The proposed method consists of three steps: data collection, statistical analysis and emotion classification. We systematically collected and analyzed the music features and corresponding emotional features by conducting a survey of various genres of existing music. Also, we derived statistical relationships for these features via multiple regression analysis based on each collected feature; we then classified the music by identifying the emotional range associated with the derived data.

This paper is organized as follows: Section 2 introduces intelligent IoT applications. Section 3 investigates related work. Section 4 explains support vector machine as background to our work. Section 5 describes the emotion-based automatic music classification method proposed in this paper. Section 6 shows the experimental design and results of the proposed method and Section 7 discusses threats to validity. Finally, Section 8 concludes this study and suggests future work.

## 2. Intelligent IoT Applications

Following the stream of the 4th Industrial Revolution, the IoT technology is becoming more and more intelligent and transforming into a high advanced stage [15]. By communicating and collaborating with each other, the IoT equipment pervades our daily lives through IoT-related services. Also by becoming more sophisticated, they are gradually realizing the future pictures we have been envisioning. As such, people's expectations and interest are rising as much as the value IoT technology is expected to create in our lives and industries.

The products embedded in IoT technology are being utilized in many ways and categories [16]. Especially, they are providing much convenience in our everyday life by becoming more intelligent over time. Global companies are investing huge research funds to develop advanced intelligent IoT applications for the areas involving healthcare, smart car, smart home and smart factory systems. For instance, they are making greater efforts to provide a more intelligent service for smart devices (e.g., smart TV, smart phone, smart pad, smartwatch, smart glass, smart speaker, etc.) which are easily accessed or widely used by general users.

In particular, since the intelligent personal assistant (IPA) [17] such as Amazon Alexa [4,5], Apple Siri [6–8], Google Assistant [9,10] or Microsoft Cortana [11,12] has been introduced, we are receiving more intelligent services. Currently, these IPA's aim to provide a wide range of services by consistently collecting various kinds of information including time, location and personal profiles. They provide a typical web search result when they are unable to reply to the user's question clearly or directly. The current IPA's basically provide information to the users in two ways, proactively or reactively [17,18]. The proactive assistant refers to the agent who automatically provides information useful to the user without his/her explicit request. That is, it recommends a specific service depending on the upcoming event or the user's current situation. The latter refers to an agent who directly responds to the question or the sentence actually entered by the user.

User's information is consistently collected, accumulated and utilized more following the development of intelligent technologies and now, aside from collecting general information, the

technologies which can provide appropriate services by identifying or recognizing user's emotion and achieve adequate interactions are being studied for commercialization. In this regard, one of the IoT-related technology fields which can easily be understood or accessed by the users is the IoT technology utilizing music. For example, certain music can be provided to the user through a smart device to satisfy his/her desire after understanding his/her emotional state or today's weather conditions.

Thus, this study attempts to focus on the R&D of a platform technology which will be able to automatically categorize the emotions associated with various types of music to support an intelligent IoT application technology which satisfies the smart device user's emotions or emotional needs and then recommend/provide proper music or musical genre accordingly.

## 3. Related Work

### 3.1. Emotional Representation

To date, there have been many studies on how to quantify emotion for emotional analysis. Consequently, various emotional models have been proposed.

First, the circumplex model by Russell, which is a well-known and frequently mentioned model in cognitive science and psychology [19–21], shows emotions on a two-dimensional coordinate plane by using two measures to represent each axis: valence and arousal. In this model, valence represents the positive and negative degree of emotion and arousal represents the intensity of emotion. The axes of emotion are drawn according to the arousal level and whether valence is positive or negative. Emotions can be arranged in a circular shape at the end of each axis or emotional maps can be drawn by extracting six major emotions. Figure 1 shows the circumplex model [19].

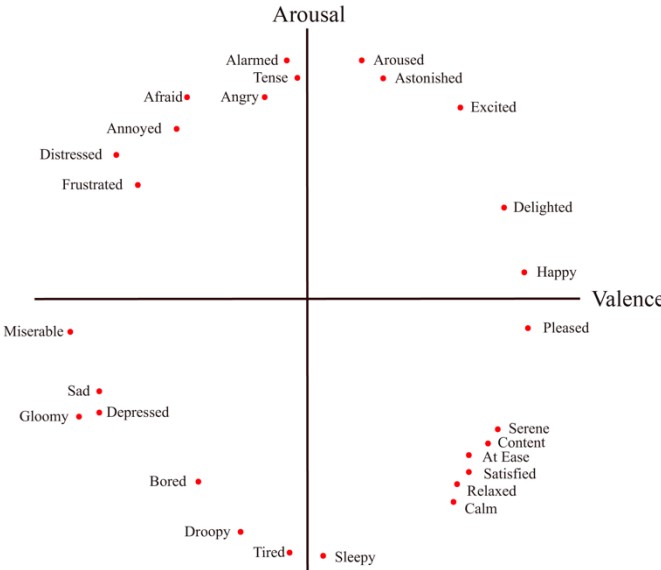

**Figure 1.** Russell's circumplex model. The circumplex model is developed by James Russell. In the model, emotions are distributed in a two-dimensional plane. The x-axis represents valence and the y-axis represents arousal. Valence refers to the positive and negative degree of emotion and arousal refers to the intensity of emotion. Using the circumplex model, emotional states can be presented at any level of valence and arousal.

Secondly, Thayer's model [22–24], which applies the circumplex model to music, implements two measures, that is, Energy and Stress corresponding to arousal and valence in the circumplex model, on the two-dimensional coordinate plane. In this model, Energy refers to the volume or intensity of sound in music and Stress refers to the tonality and tempo of music. According to the level of stress

and energy, music mood can be divided into four clusters: Exuberance, Anxious, Contentment and Depression. Figure 2 shows the Thayer's model [25].

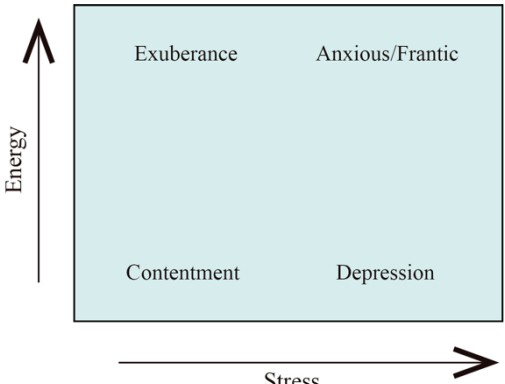

**Figure 2.** Thayer's model. The Thayer's model is proposed by Robert E. Thayer. In the model, emotional states are represented by energy and stress corresponding to arousal and valence of the circumplex model. Energy refers to the volume or intensity of sound in music and Stress refers to the tonality and tempo of music. The mood is divided into four clusters: calm-energy (e.g., Exuberance), calm-tiredness (e.g., Contentment), tense-energy (e.g., Frantic) and tense-tiredness (e.g., depression). Using the Thayer's model, it is possible to characterize musical passages according to the dimensions of energy and stress.

The third model, developed by Tellegen, Watson and Clark [26,27], uses positive/negative affect, pleasantness/unpleasantness and engagement/disengagement to classify a greater variety of emotions than previous models. Figure 3 shows the Tellegen-Watson-Clark model [26].

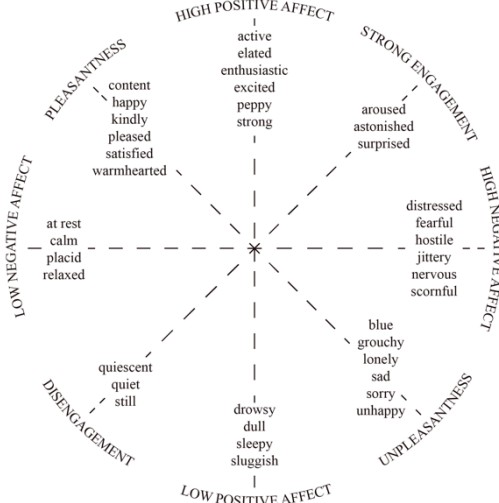

**Figure 3.** Tellegen-Watson-Clark model. In the model, high positive affect and high negative affect are proposed as two independent dimensions, whereas pleasantness–unpleasantness and engagement-disengagement represent the endpoints of one bipolar dimension.

The main advantage of representing emotion in a dimensional form is that any emotion can be mapped in that space. It allows a model where any emotion can be represented, within the limitation of these dimensions.

*3.2. Music Emotion Recognition*

Music emotion recognition is a field that aims to study the relationship between music and emotion and is helpful in music understanding and music information retrieval. The related studies generally focuses on recognizing the internal emotion of given music.

Patra et al. [28] proposed the music emotion recognition system to find the dynamic valence and arousal values of a song. They adopted the feed-forward neural network with 10 hidden layers to build the regression model and used the correlation-based method to find out suitable features. Chen et al. [29] proposed a system for detecting emotion in music that is based on a deep Gaussian process. The proposed system consisted of two major parts: feature extraction and classification. In the classification part, deep Gaussian process was utilized for emotion recognition. Deep Gaussian process provided structural learning in Gaussian process model. Accordingly, the potential and efficiency of learning methodologies with deep architecture were proved in statistical machine learning. Bargaje et al. [30] proposed an approach for emotion detection from audio samples by optimizing the feature set to reduce the computation time using the genetic algorithm. In particular, 2D (Arousal-Valence) type plane was modified into 3D (Arousal-Valence-Loudness) plane to identify eight emotions. Maik et al. [31] studied a method consisting of stacked convolutional and recurrent neural networks for continuous prediction of emotion in two-dimensional Valence-Arousal space. In the study, they utilized one CNN layer followed by two branches of RNNs trained separately for valence and arousal. Ascalon et al. [32] focused on recognizing music mood using lyrics. While most researches employed audio alone or using two or more sources of features for their study, this work tried to investigate the relationship between lyrics and mood. Word level features such as TF-IDF (Term Frequency-Inverse Document Frequency) and keyGraph keyword generation algorithm were applied using different thresholds and parameters. For each song, the emotion was then categorized as one of the five following emotions: 'Saddest,' 'Sad,' 'Neutral,' 'Happy' or 'Happiest.'

Generally, based on the various acoustic features of music, many studies used different classifier to recognize music emotion, such as K-nearest neighbors [33], support vector regression [34] and support vector machines [35,36]. Unlike the previous studies, in our study, new music features were identified by discussing with the working group for developing intelligent IoT applications in a small and medium-sized enterprise. Also, a combined approach with regression and support vector machine was applied. Moreover, through the survey, induced (evoked) emotion was used for the validation of the proposed method. Note that, according to the studies [37,38], two types of emotions can be occurred when listening to music: (1) perceived emotions (emotions that are communicated by the music source) and (2) induced emotions (emotional reaction that the music source provokes in listeners). Because the perceived emotion is the emotion that the source conveys, the perceived emotion of happy songs is always "happy." However, because the induced emotion is more subjective, the same happy songs may not necessarily evoke happiness in the listeners. The previous studies tried to recognize the perceived emotion of songs since it is relatively objective and invariant to the context (mood, environment, etc.). However, this may not be practical because the emotion is originally derived from the feeling of humans. Thus, in our study, we validated the proposed method based on the induced emotion of music collected from the survey participants.

## 4. Background

Support Vector Machine (SVM) [39,40] is a machine learning algorithm that recognizes patterns in large data sets and classifies each data as a class. Given a set of data belonging to either of two classes, the SVM algorithm generates a non-stochastic binary linear classification model that determines to which class the new data belongs based on a given set of data. The generated classification model is represented by boundaries in the space where data is mapped and the SVM algorithm finds the boundary with the largest width. For example, it finds a maximum-margin hyperplane with the largest margin that is the shortest distance between the data vector and the hyperplane, which is among the hyperplanes that divide the data vectors when classifying two classes of data vectors A and B, as is

shown in Figure 4. Here, the data vector having the shortest distance to the hyperplane is called a support vector.

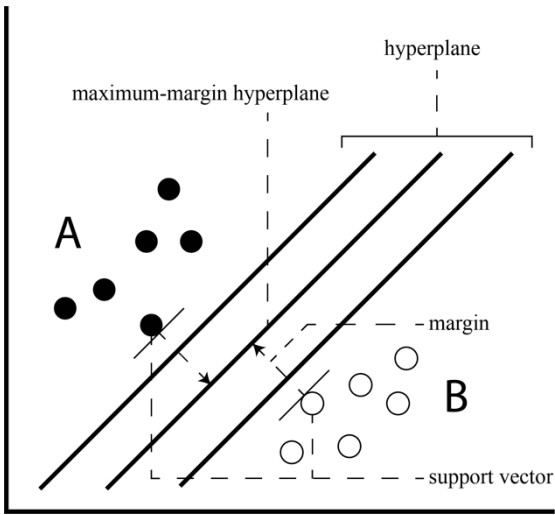

**Figure 4.** Hyperplane and margin of SVM.

This study uses SVM to generate a classification model to determine which type of emotional data associated with a particular song belong to a specific data set.

## 5. Overall Approach

In order to classify emotions according to music, we conduct an emotional identification survey for music and also extract the characteristic data from the music and then regression analysis is applied based on the collected data. Finally, we determine emotions associated with music based on an emotion classification map derived from SVM. Figure 5 provides an illustration of the emotion-based automatic music classification method.

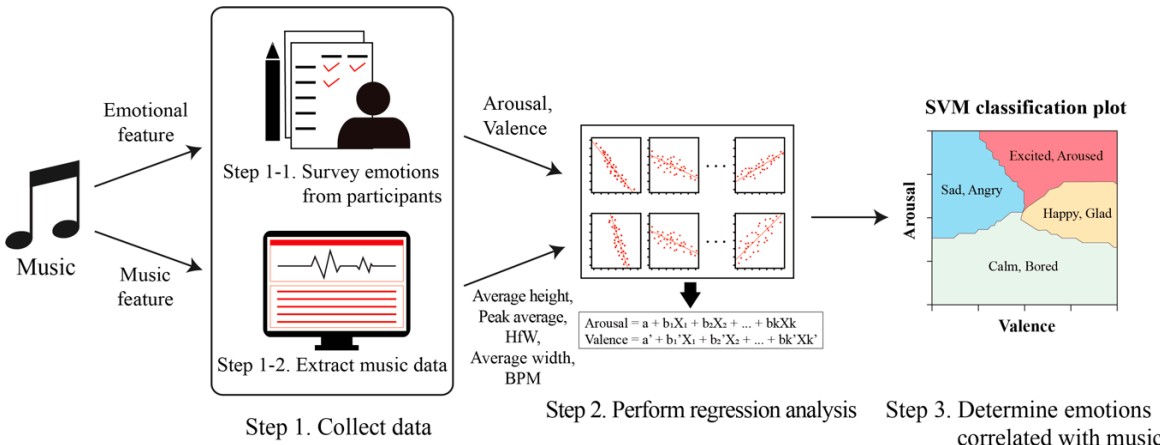

**Figure 5.** Overall approach. The approach consists of three steps: (1) emotional and music data are collected from participants and songs, (2) regression analysis is applied to identify the relationship between emotional data and music data and (3) a classification algorithm is applied to determine how the emotional data correlated with the music data are related to the actual emotions.

### 5.1. Step 1: Collect Data

In Step 1, we measure emotional data associated with each song by carrying out a survey and also collect the characteristic data, such as the amplitude and frequency of the music. Note that there are

cases in which various moods are included throughout one song. In this case, it is not easy to conclude as one emotion for an entire song. Thus, emotional data are measured by decomposing each song into 20-s sections to collect more detailed and accurate data; the characteristic data are also analyzed over intervals of 20 seconds.

### 5.1.1. Step 1-1: Survey Emotions from Participants

People experience countless emotions such as joy, excitement, anger and sadness while listening to music and different people feel different emotions for the same song. Furthermore, the emotions associated with music are very subjective, so it is very difficult to objectively measure them. Therefore, when we conduct a survey on the emotions associated with music, we use the general well-known circumplex emotional model [15] and examine the emotions of participants according to two measures: valence and arousal.

### 5.1.2. Step 1-2: Extract Music Data

Sound has an intensity, pitch and tone, which can be interpreted as amplitude, frequency and waveform, respectively, when physically analyzing them. Thus, these characteristic data are collected as music data.

When analyzing the characteristics of music, (1) Average height, (2) Peak average, (3) HfW (the number of half wavelengths), (4) Average width and (5) BPM (beats per minute) of each song are extracted. As is shown in Figure 6, Average height refers to the mean height of all the points of the midline and Peak average refers to the mean maximum height of the waves. HfW refers to the number of positive sections in which the amplitude of a wave increases from zero to a positive value and subsequently decreases from a positive value to zero and negative sections in which the amplitude of a wave decreases from 0 to a negative value and subsequently increases from a negative value to zero. Average width refers to the mean width of such sections. BPM refers to the tempo that is a musical term for the pace or speed of a piece of music.

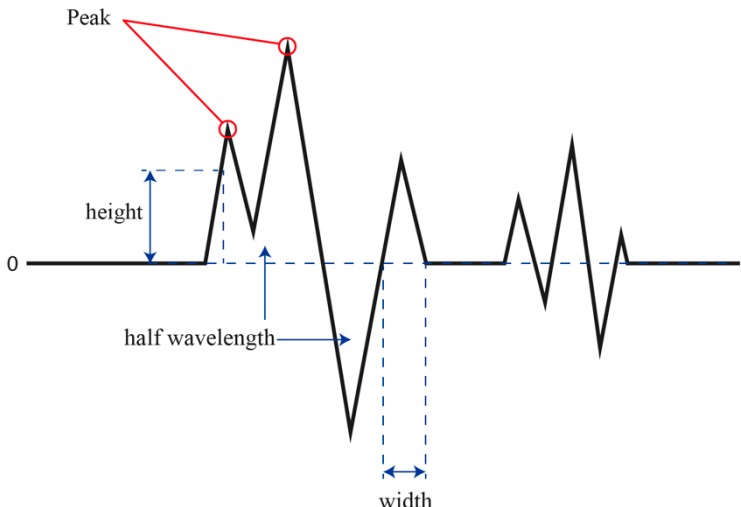

**Figure 6.** Example of a music wave data section.

### *5.2. Step 2: Perform Regression Analysis*

Music is composed of various elements such as rhythm, tone and tempo. It is not difficult for a person to determine the mood of music by identifying these features. However, it is a very difficult task to determine which emotions are associated with the particular music by only examining the characteristics of music, such as the average amplitude and wave number.

In Step 2, regression analysis is used to infer which emotions a person feels when listening to certain music via analysis of the characteristics of the music. This is performed with the aim of correlating the emotions of the person to the music data by analyzing the correlation between the valence and arousal values obtained via Step 1-1 and the results of music data analysis obtained via Step 1-2.

Average height, Peak average, HfW, Average width and BPM are implemented as the independent variables in the regression analysis. Valence and arousal data collected from the participants are used as the dependent variables. In our study, stepwise regression is used to determine the best-fitting set of independent variables contributing to each dependent variable [41]. Independent variables are considered to be significant predictors if $p < 0.05$.

*5.3. Step 3: Determine Emotions Correlated with Music*

Step 3 entails performing the task of determining how the valence and arousal values correlated with the music data obtained via regression analysis (Step 2) are related to the actual emotions.

As is shown in Figure 7, the circumplex model [19] is employed to classify the emotions into four emotional categories according to the valence and arousal values of the 20-s interval of music; the four emotional categories are as follows: 'Happy, Glad,' 'Excited, Aroused,' 'Sad, Angry' and 'Calm, Bored.'

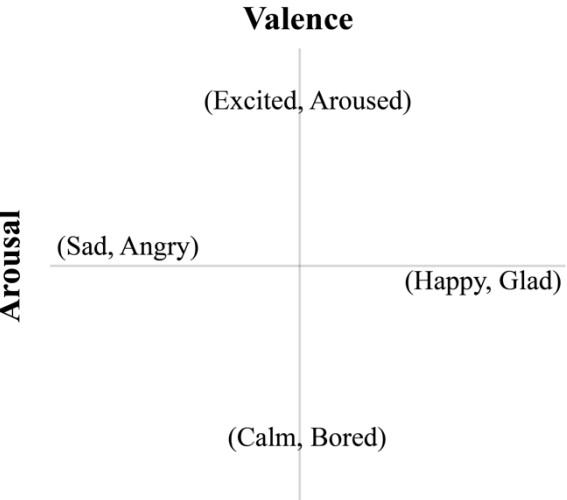

**Figure 7.** Classified emotional categories.

The corresponding quantitative emotional values (i.e., valence and arousal) determined for each person are displayed on the plane shown in Figure 7; the value data is partitioned into each category via SVM [39,40]. This allows the emotional values to be mapped to a particular actual emotional category.

Thus, when classifying a new (unlearned) song as based on the proposed method, the emotions for the song are automatically classified by mapping the emotional values (valence and arousal) resulting from regression analysis into the emotional categories that have been identified via SVM, after the characteristic data of the new song were extracted.

## 6. Experiment

*6.1. Experimental Design*

In order to investigate the emotions that people feel when listening to music, we recruited 40 participants hoping to join this experiment; 31 participants in their 20s, 6 participants in their 30s and 3 participants in their 40s. In this study, the music files used to collect data for the survey were 100 Korean pop songs that gained global popularity. Various genres such as dance, ballad, R&B, Hip-Hop and rock were selected. The audio source was an MP3 file with a sound quality of 320 kbps.

Please note that there are a wide variety of musical genres and even the same music could be perceived depending on the person. Although the experiments conducted in this study have focused on Korean pop songs (K-pops) and as it is possible that those who are addicted to classical or heavy metal music genre would not enjoy them at all, they would feel different emotions. The main reason for choosing K-pops is that they are not only receiving global attention and becoming more popular [42] but also embrace various musical genres. K-pops were used for the study as the music in each K-pop genre allows individual listeners to feel a quite specific emotion and considered to be appropriate for the evaluation of emotion identification performances.

For Step 1-1, an examination into emotions was conducted by surveying each participant listening to above 100 songs; the form of the surveys is shown in Table 1. The form covered items such as music title, time, valence, arousal and emotion. Music title refers to the title of the song and time refers to a 20-s interval section of the song. Valence and arousal are the participant's valence and arousal values associated with the music; these values range from $-100$ to 100. Emotion is the perception of which emotion is felt by the participant in each 20-s interval section. The Emotion item allows them to select four categories, that is, 'Happy, Glad,' 'Excited, Aroused,' 'Sad, Angry' and 'Calm, Bored.' We provided a detailed explanation of the survey before conducting it to ensure that the participants could best provide accurate and objective data, as emotion is very subjective and they were not familiar with the circumplex model.

**Table 1.** Example of the form of the surveys.

| Music title | Time (s) | Valence | Arousal | Emotion | | | |
|---|---|---|---|---|---|---|---|
| | | | | Happy, Glad | Excited, Aroused | Sad, Angry | Calm, Bored |
| A | ~ 20 | −63 | −88 | | | | ✓ |
| | ~ 40 | −50 | −83 | | | | ✓ |
| | ~ 60 | −25 | −81 | | | | ✓ |
| | ~ 80 | −8 | −78 | | | ✓ | |
| | ~ 100 | −5 | −75 | | | ✓ | |
| | ~ 120 | 5 | −75 | | | ✓ | |
| B | ~ 20 | 20 | 35 | ✓ | | | |
| | ~ 40 | 22 | 30 | ✓ | | | |
| | ~ 60 | 35 | 30 | | ✓ | | |
| | ~ 80 | 32 | 30 | | ✓ | | |
| | ~ 100 | 28 | 20 | ✓ | | | |
| | ~ 120 | 28 | 20 | ✓ | | | |

For Step 1-2, the extraction and analysis tool for the features of music were directly implemented using Java programming language. To calculate the BPM, we used the Easy BPM Calculation in Java, which is open-source software [43]. The music data were extracted in the format shown in Table 2 by utilizing these automation tools.

**Table 2.** Collected music data.

| Music title | Time (s) | Average height | Peak average | HfW | Average width | BPM |
|---|---|---|---|---|---|---|
| A | ~ 20 | 4421 | 4848 | 247 | 557 | 59 |
| | ~ 40 | 8059 | 8890 | 260 | 482 | 59 |
| | ~ 60 | 5631 | 6067 | 317 | 428 | 59 |
| | ~ 80 | 4900 | 5249 | 258 | 503 | 59 |
| | ~ 100 | 7301 | 8067 | 250 | 497 | 59 |
| | ~ 120 | 8774 | 9694 | 272 | 455 | 59 |
| B | ~ 20 | 3547 | 4010 | 377 | 385 | 95 |
| | ~ 40 | 5350 | 5828 | 252 | 513 | 95 |
| | ~ 60 | 4657 | 5144 | 242 | 581 | 95 |
| | ~ 80 | 6813 | 7264 | 263 | 490 | 95 |
| | ~ 100 | 5718 | 5769 | 178 | 757 | 95 |
| | ~ 120 | 6765 | 7043 | 180 | 754 | 95 |

For Step 2, by using the data obtained via emotional examination and music data extraction, we first removed the outliers with extreme values before performing regression analysis to increase the reliability of the analysis. Subsequently, regression analysis was conducted by using the statistical computing tool R [44] applying valence and arousal as dependent variables and Average height, Peak average, HfW, Average width and BPM as independent variables. Figures 8 and 9 show the results of the simple regression analysis of each independent variable for dependent variables. Through the stepwise regression procedure, Average height, Peak average and BPM were selected as the valid independent variables for valence; Average height, Peak average, HfW and Average width were selected as the valid independent variables for arousal. The models for valence and arousal were derived as follows:

$$\text{Valence} = (-209.7) + (0.004548 * \text{Average height}) + (0.0005603 * \text{Peak average}) + (2.29 * \text{BPM}) \quad (1)$$

$$\text{Arousal} = (-76.228036) + (0.012440 * \text{Average height}) + (-0.006034 * \text{Peak average}) + (0.240159 * \text{HfW}) + (-0.005287 * \text{Average width}) \quad (2)$$

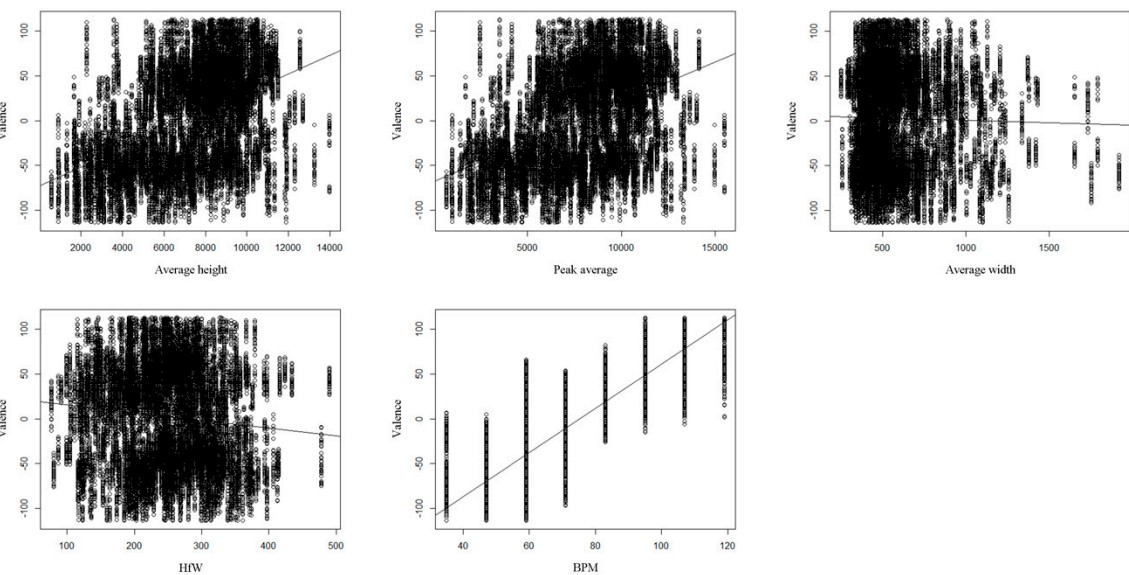

**Figure 8.** Data distribution for valence with each independent variable.

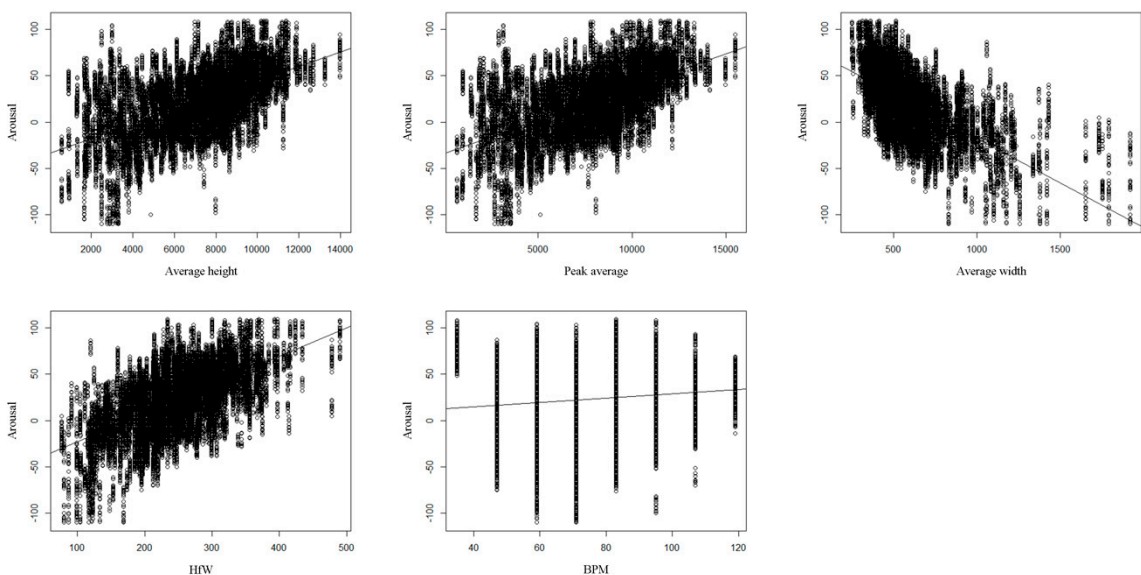

**Figure 9.** Data distribution for arousal with each independent variable.

For Step 3, as is shown in Figure 10, the "Emotion Classification Map" classified by the four emotional categories was derived by using SVM, based on the results obtained via survey participants. The parameter setting is as follows: Kernel is linear, Tolerance is 0.001 (tolerance of termination criterion) and epsilon is 0.1 (epsilon in the insensitive-loss function). Using the classification algorithm, SVM, it is possible to automatically classify emotions associated with new (unlearned) songs by mapping the results obtained via multiple regression analysis to an emotion classification map.

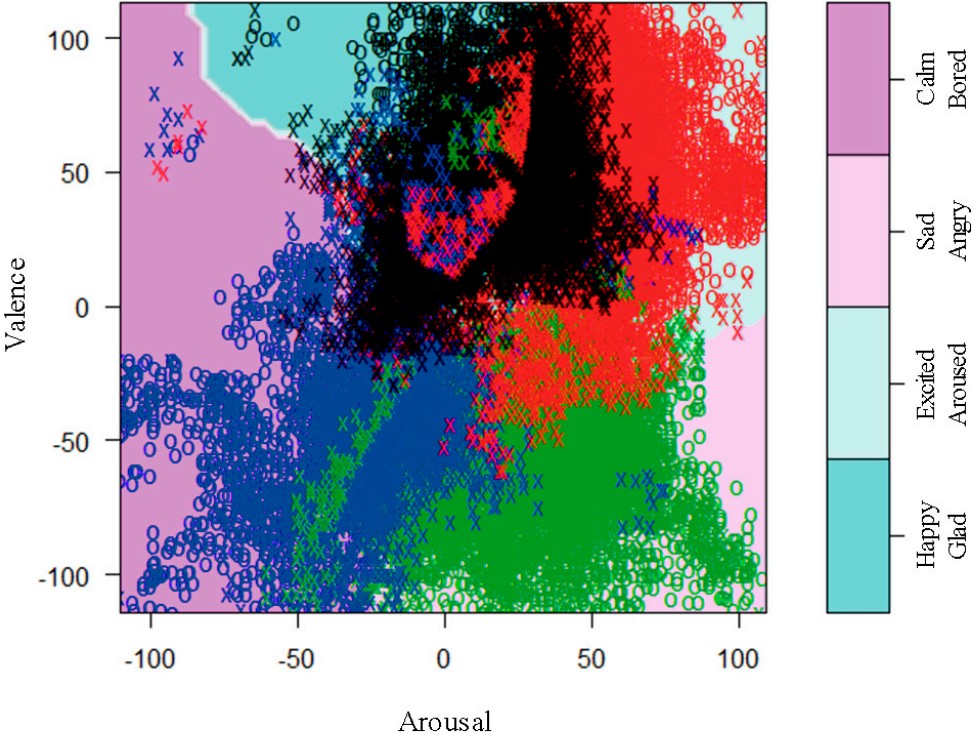

**Figure 10.** Emotion Classification Map.

## 6.2. Experimental Results

In order to verify the effectiveness of the proposed method, a total of 35 new Korean pop songs were used as the testing set. The results of the emotion classification investigation of 35 songs were obtained by surveying 40 participants and evaluated with respect to the emotional category derived via the proposed method for each song.

Table 3 shows the probability (%) of occurrence of the same emotion by comparing the results from the proposed method to the results from the survey participants, in each 20-s interval section. In Table 3, TPM (The Proposed Method) refers to the method proposed in this study and A to AN are the survey participants. The numbers in each column show the emotion match rate between the results obtained via TPM and those obtained via the survey participants. AVG is the average value of the emotion match rate for TPM and each participant. The bold numbers indicate the best emotion match rate between TPM and each participant. All values are rounded off to 3 decimal places.

**Table 3.** Emotion match rate.

| (%) | | Comparison of TPM and Survey Participants (A to AN, 40 Participants) | | | | | | | | | | | | |
|---|---|---|---|---|---|---|---|---|---|---|---|---|---|---|
| | A | 72.94 | B | 74.71 | C | 74.12 | D | 74.71 | E | 75.88 | F | 74.12 |
| | G | 72.94 | H | 72.94 | I | 71.76 | J | 72.94 | K | 75.29 | L | 72.94 |
| | M | 74.71 | N | 74.12 | O | 73.53 | P | 72.35 | Q | 75.88 | R | **77.65** |
| TPM | S | 72.35 | T | 72.35 | U | 72.94 | V | 71.76 | W | 71.18 | X | 75.88 |
| | Y | 76.47 | Z | **77.65** | AA | 72.35 | AB | 72.94 | AC | 75.29 | AD | 74.71 |
| | AE | 74.71 | AF | 70.59 | AG | 74.12 | AH | 72.94 | AI | 73.53 | AJ | 74.12 |
| | AK | 75.88 | AL | 75.29 | AM | 75.29 | AN | 72.35 | | AVG: 73.96 | | |

As is shown in Table 3, the average match rate was 73.96% and the maximum match rate was 77.65% obtained from R and Z. The match rates obtained via surveying participants were found to range from 70.59% to 77.65%.

For more comprehensive analysis, the emotion match rates were compared by classifying them from the perspective of three different cases: "All," "Most" and "Least." Figure 11 illustrates the comparison approaches in detail. In Figure 11, Songs refers to the 35 new songs used as the testing set. First, the case "All" means that the emotional results derived via TPM and the emotional results from the survey all match with one emotion. Secondly, the case "Most" means that the emotion classification obtained via TPM is in agreement with the most commonly classified emotion among the survey results. As the third case "Least" means that the emotion classification obtained via TPM exists at least once among the survey results. Thus, "Most" includes "All" and "Least" includes "Most" and "All" as presented in Figure 11.

**Figure 11.** Comparison approaches.

Based on the three cases, Table 4 shows the emotion match rates for TPM and each participant. The first row in Table 4 identifies the TPM and each participant. The second, third and fourth rows indicate the match rate for "All," "Most" and "Least." The bold numbers indicate the performance of the emotion match rate via TPM.

**Table 4.** Emotion match rates for the "All," "Most" and "Least" cases.

| (%) | TPM | A | B | C | D | E | F |
|---|---|---|---|---|---|---|---|
| All | **44.71** | 44.71 | 44.71 | 44.71 | 44.71 | 44.71 | 44.71 |
| Most | **77.06** | 87.06 | 87.06 | 87.65 | 90.00 | 82.35 | 87.65 |
| Least | **94.12** | 100.00 | 100.00 | 100.00 | 100.00 | 100.00 | 100.00 |

| | G | H | I | J | K | L | M |
|---|---|---|---|---|---|---|---|
| All | 44.71 | 44.71 | 44.71 | 44.71 | 44.71 | 44.71 | 44.71 |
| Most | 85.88 | 87.06 | 88.24 | 92.35 | 88.82 | 86.47 | 90.00 |
| Least | 100.00 | 100.00 | 100.00 | 100.00 | 100.00 | 100.00 | 100.00 |

| | N | O | P | Q | R | S | T |
|---|---|---|---|---|---|---|---|
| All | 44.71 | 44.71 | 44.71 | 44.71 | 44.71 | 44.71 | 44.71 |
| Most | 85.88 | 85.88 | 90.00 | 90.00 | 89.41 | 88.82 | 86.47 |
| Least | 100.00 | 100.00 | 100.00 | 100.00 | 100.00 | 100.00 | 100.00 |

| | U | V | W | X | Y | Z | AA |
|---|---|---|---|---|---|---|---|
| All | 44.71 | 44.71 | 44.71 | 44.71 | 44.71 | 44.71 | 44.71 |
| Most | 88.82 | 90.00 | 85.88 | 88.82 | 86.47 | 85.29 | 84.71 |
| Least | 100.00 | 100.00 | 100.00 | 100.00 | 100.00 | 100.00 | 100.00 |

| | AB | AC | AD | AE | AF | AG | AH |
|---|---|---|---|---|---|---|---|
| All | 44.71 | 44.71 | 44.71 | 44.71 | 44.71 | 44.71 | 44.71 |
| Most | 87.65 | 90.00 | 88.82 | 89.41 | 89.41 | 88.24 | 87.06 |
| Least | 100.00 | 100.00 | 100.00 | 100.00 | 100.00 | 100.00 | 100.00 |

| | AI | AJ | AK | AL | AM | AN | - |
|---|---|---|---|---|---|---|---|
| All | 44.71 | 44.71 | 44.71 | 44.71 | 44.71 | 44.71 | - |
| Most | 83.53 | 88.82 | 87.65 | 88.24 | 85.29 | 90.59 | - |
| Least | 100.00 | 100.00 | 100.00 | 100.00 | 100.00 | 100.00 | - |

As presented in Table 4, in the case of "All," the emotional match rate was the same as 44.71% for TPM and each participant. Because it was an uncommon occurrence for all of the survey participants to always feel the same emotion each other, the match rate was not high. Note that, in each song, there was no case where only the emotional result derived via TPM was different when the emotional results derived via survey participants were all the same as one emotional type. That is, the low match rate was caused by one or more different emotional results via survey participants. Because emotion perception is essentially subjective, one could perceive different emotions even when listening to the same song. Thus, the following "Most" could be more valid experimentation strategy. In the case of "Most," the match rate via TPM was 77.06%. That is, TPM could identify fairly large of the emotions felt by the participants (the emotion classification results obtained via TPM are in agreement at a rate of 77.06% with the most commonly classified emotions among the results of the survey). In the case of "Least," the match rate via TPM was 94.12%. That is, TPM could identify at least one of the emotions that were felt by many participants (the emotion classification results obtained via TPM are in agreement at a rate of 94.12% with one of the emotions among the results of the survey).

Figure 12 provides the overall results of the emotion match rates for "All," "Most" and "Least." The experimental results according to each of the three perspectives were comprehensively compared and analyzed. In Figure 12, Survey participants represent the average of the emotion match rates of the survey participants and TPM represents the emotion match rate determined via the method proposed in this study.

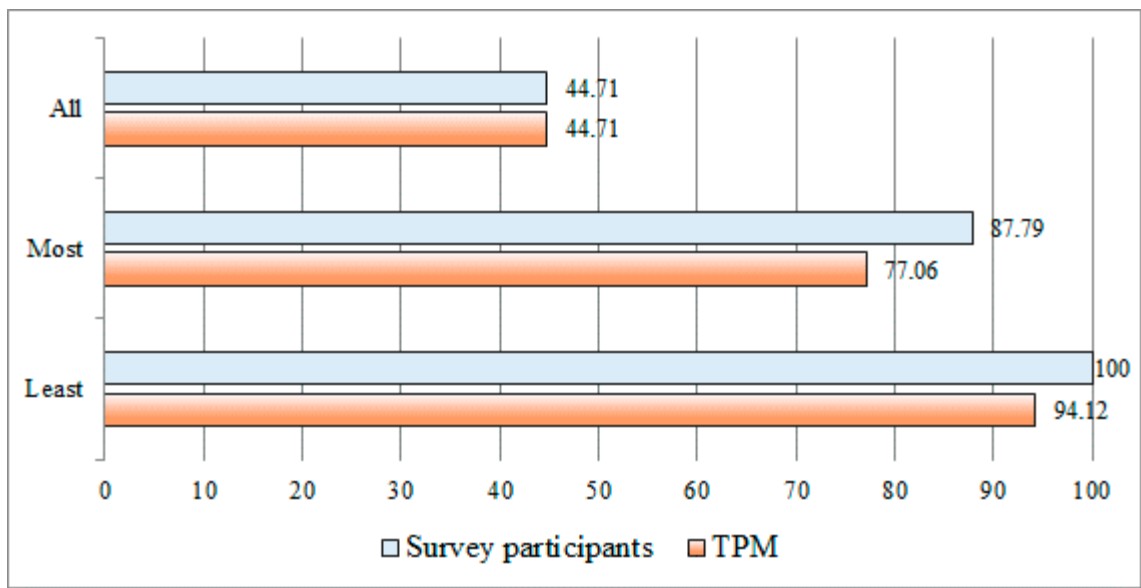

**Figure 12.** Comparison of the emotion match rates.

For the three cases, as presented in Figure 12, the emotion match rates obtained via TPM were 44.71%, 77.06% and 94.12%, whereas the average emotion match rates from the survey were 44.71%, 87.79% and 100%, respectively. These results demonstrate that the emotions via the survey participants are largely identified via TPM. Of these cases, the match rate via TPM from the perspective of "Most" was 77.06%, which is approximately 10% lower than the 87.79% average obtained via the survey participants. Although the rate of TPM is a little lower than that of the survey participants, this confirms that TPM can be effectively utilized as an automatic method to identify and classify the common emotions felt by people when they listen to songs.

*6.3. Comparative analysis with different classification algorithms*

In the proposed method, SVM is used to identify the emotional categories and classify the emotions for the music. However, as is well known, there are several different classification algorithms aside from SVM. Thus, in our experiment, we additionally analyze the proposed method with popular classification algorithms, such as random forest [45], deep neural network [46] and K-nearest neighbor [33,46], as well as SVM. The configuration and parameter sets are as follows:

- SVM: Kernel is linear, Tolerance is 0.001 (tolerance of termination criterion) and epsilon is 0.1 (epsilon in the insensitive-loss function)
- Random forest: the coefficient(s) of regularization is 0.8 and regularization is true (1)
- Deep neural network: hidden dim is 30 (the dimension of each layer), learning rate is 0.1, epoch num is 1000 (1000 iteration over the entire dataset) and hidden layer is 3
- K-nearest neighbor: K is set from 2 to 6

The overall experimental results are shown in Table 5. In Table 5, the first column indicates the survey participants and the other columns indicate the experimental results from classification algorithms. RF means random forest, DNN means deep neural network and KNN denotes K-nearest neighbor. The bold and underlined numbers indicate the best emotion match rate in each participant. All values are rounded off to 3 decimal places.

**Table 5.** Comparative analysis for emotion match rate with classification algorithms.

| Participants | SVM | RF | DNN | KNN | | | | |
|:---:|:---:|:---:|:---:|:---:|:---:|:---:|:---:|:---:|
| | | | | K = 2 | K = 3 | K = 4 | K = 5 | K = 6 |
| A | **72.94** | 68.24 | 71.76 | 64.71 | 67.65 | 68.82 | 69.41 | 68.24 |
| B | **74.71** | 70.00 | 73.53 | 65.88 | 68.24 | 69.41 | 71.18 | 67.06 |
| C | **74.12** | 68.24 | 71.76 | 62.35 | 67.65 | 68.24 | 69.41 | 67.06 |
| D | **74.71** | 68.82 | 74.71 | 62.94 | 68.24 | 69.41 | 70.00 | 67.65 |
| E | **75.88** | **75.88** | 72.35 | 70.00 | 72.94 | 72.94 | 72.35 | 71.18 |
| F | 74.12 | 69.41 | **74.71** | 63.53 | 67.06 | 68.82 | 71.18 | 68.24 |
| G | **72.94** | 69.41 | 72.35 | 64.12 | 68.24 | 68.82 | 70.00 | 68.24 |
| H | **72.94** | 67.06 | **72.94** | 64.71 | 67.65 | 69.41 | 69.41 | 65.88 |
| I | **71.76** | 67.06 | 70.59 | 60.59 | 64.71 | 65.88 | 67.65 | 65.29 |
| J | **72.94** | 68.24 | 71.76 | 62.94 | 66.47 | 68.82 | 70.59 | 65.88 |
| K | 75.29 | 70.59 | **76.47** | 66.47 | 71.18 | 70.59 | 70.59 | 69.41 |
| L | 72.94 | 67.06 | **74.12** | 63.53 | 66.47 | 69.41 | 69.41 | 67.06 |
| M | **74.71** | 70.00 | 73.53 | 65.88 | 70.59 | 71.76 | 73.53 | 68.82 |
| N | **74.12** | 70.59 | 73.53 | 65.88 | 70.00 | 70.00 | 70.59 | 68.24 |
| O | **73.53** | 67.65 | 72.35 | 62.35 | 67.06 | 68.24 | 68.82 | 67.65 |
| P | 72.35 | 68.82 | **72.94** | 64.71 | 67.65 | 70.59 | **72.94** | 67.06 |
| Q | 75.88 | 70.00 | **77.06** | 67.06 | 70.00 | 72.94 | 72.94 | 72.35 |
| R | **77.65** | 72.94 | 73.53 | 65.88 | 69.41 | 71.18 | 71.18 | 68.24 |
| S | **72.35** | 65.29 | 71.18 | 60.59 | 64.71 | 67.06 | 68.82 | 65.29 |
| T | **72.35** | 66.47 | 70.59 | 62.35 | 65.88 | 66.47 | 66.47 | 66.47 |
| U | **72.94** | 68.24 | 68.82 | 63.53 | 67.65 | 67.65 | 68.24 | 64.71 |
| V | 71.76 | 68.24 | **74.12** | 64.12 | 66.47 | 68.82 | 69.41 | 67.06 |
| W | 71.18 | 67.65 | **72.94** | 62.35 | 66.47 | 67.06 | 69.41 | 67.06 |
| X | **75.88** | 70.00 | 74.71 | 62.94 | 68.24 | 69.41 | 70.00 | 68.82 |
| Y | 76.47 | 71.76 | **77.06** | 67.65 | 71.76 | 71.18 | 72.35 | 71.76 |
| Z | **77.65** | 70.59 | 75.88 | 67.06 | 71.18 | 71.18 | 72.94 | 71.76 |
| AA | **72.35** | 67.65 | 70.59 | 62.35 | 65.88 | 68.24 | 68.82 | 65.29 |
| AB | **72.94** | 68.24 | **72.94** | 62.94 | 67.65 | 67.65 | 68.24 | 67.06 |
| AC | **75.29** | 67.06 | 72.35 | 62.94 | 68.82 | 69.41 | 69.41 | 68.24 |
| AD | **74.71** | 70.00 | 73.53 | 66.47 | 69.41 | 70.59 | 71.18 | 68.82 |
| AE | **74.71** | 70.00 | 72.94 | 63.53 | 68.24 | 69.41 | 70.00 | 67.65 |
| AF | **70.59** | 65.88 | 69.41 | 61.18 | 64.71 | 65.88 | 66.47 | 64.12 |
| AG | **74.12** | 71.76 | **74.12** | 64.71 | 70.00 | 70.00 | 71.76 | 68.24 |
| AH | **72.94** | 67.06 | 71.76 | 62.35 | 66.47 | 68.24 | 68.24 | 64.71 |
| AI | **73.53** | 67.65 | 72.35 | 62.94 | 68.82 | 67.06 | 68.24 | 67.06 |
| AJ | **74.12** | 69.41 | 71.76 | 63.53 | 67.65 | 68.82 | 70.59 | 65.88 |
| AK | **75.88** | 71.18 | 74.71 | 65.88 | 69.41 | 70.59 | 72.35 | 70.00 |
| AL | **75.29** | 68.24 | 71.76 | 64.12 | 68.24 | 68.82 | 70.00 | 68.24 |
| AM | **75.29** | 70.59 | 72.35 | 64.71 | 69.41 | 70.00 | 71.18 | 68.82 |
| AN | **72.35** | 67.65 | 70.00 | 62.94 | 65.88 | 68.82 | 70.00 | 65.29 |
| **Average** | **73.96** | **69.01** | **72.90** | **64.12** | **68.10** | **69.19** | **70.13** | **67.65** |

As presented in Table 5, for all of the survey participants, 32 out of 40 participants showed the highest emotion match rate when using SVM. Next, 11 out of 40 participants presented the best emotion match rate when using DNN. RF and KNN (K = 5) showed the best rate for the 1 participant (E and P).

Figure 13 provides the average match rate of the proposed method with the various classification algorithms. The best average match rate was 73.96% obtained via SVM and the second one was 72.90% obtained via DNN. The lowest average match rate was 64.12% via KNN (K = 2).

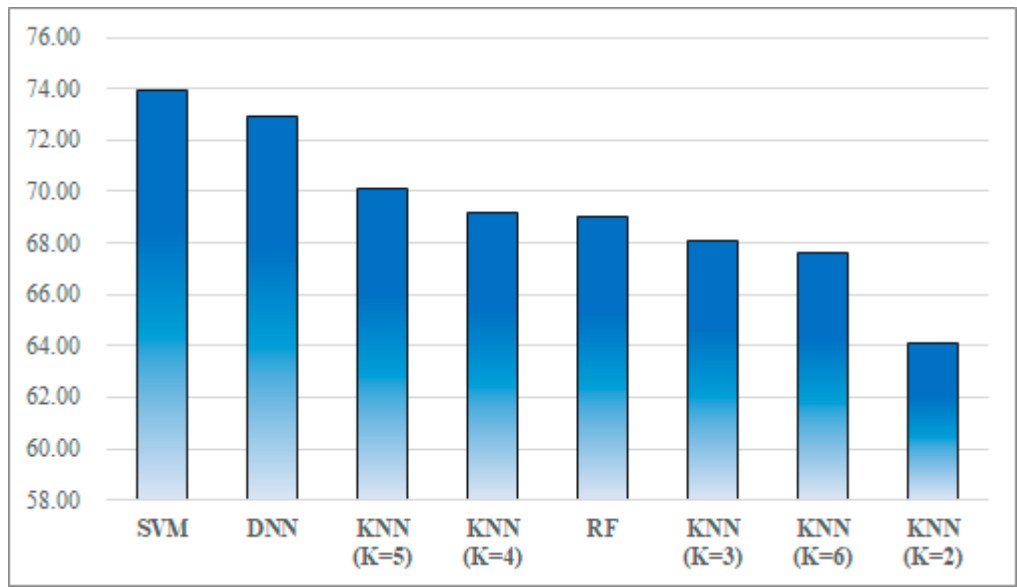

**Figure 13.** Average match rate from classification algorithms.

Table 6 shows the emotion match rate according to "All," "Most," "Least" with the classification algorithms. The bold and underlined numbers indicate the best emotion match rate for each perspective. As shown in Table 6, compared with the other classification algorithms, SVM showed a better performance for each of the three perspectives.

**Table 6.** Emotion match rate for the "All," "Most" and "Least" cases with classification algorithms.

| (%) | SVM | RF | DNN | KNN | | | | |
| --- | --- | --- | --- | --- | --- | --- | --- | --- |
| | | | | K = 2 | K = 3 | K = 4 | K = 5 | K = 6 |
| **All** | **44.71** | 41.18 | 43.53 | 38.24 | 41.18 | 41.18 | 42.35 | 40.00 |
| **Most** | **77.06** | 72.35 | 74.71 | 66.47 | 70.59 | 71.76 | 72.35 | 70.00 |
| **Least** | **94.12** | 90.59 | 94.12 | 86.47 | 89.41 | 90.00 | 90.59 | 88.82 |

## 7. Threats to Validity

One thing to keep in mind when considering the results of this experiment is that emotions can vary depending on the sound quality of music. Figure 14 shows the waveform when a song (S) has a high-quality sound and Figure 15 shows the waveform when the same portion of the same song has a low-quality sound. The results of applying the method presented in this study showed that 'Excited, Aroused' was derived for the high-quality sound version of S, whereas 'Sad, Angry' was identified for the low-quality sound version. It can be seen that the low-quality version may not produce valid results, as most survey participants identified their emotional reaction to S as 'Excited, Aroused.' Experiments in this study were performed using the high-quality version of song files (320 kbps). This is a problem in these types of experiments that must be acknowledged, as the low-quality version of song files can yield significantly different waveforms as compared to the high-quality version.

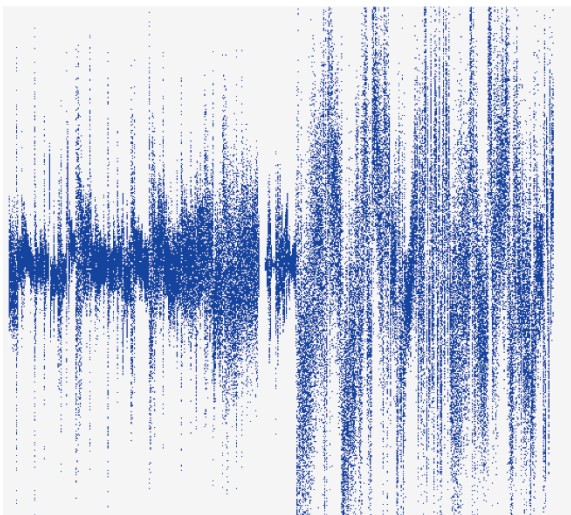

**Figure 14.** Waveform of the high-quality version of S.

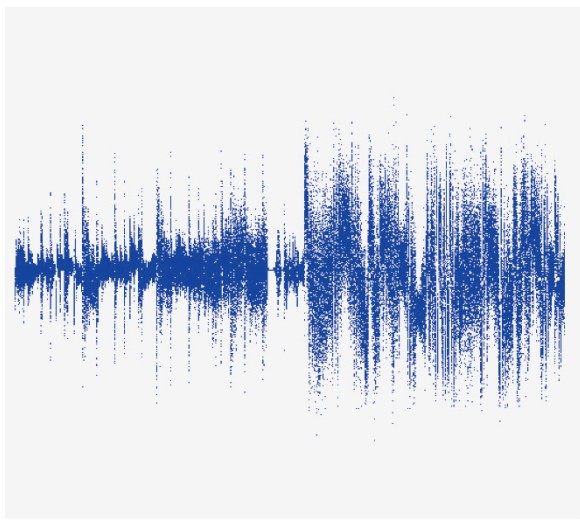

**Figure 15.** Waveform of the low-quality version of S.

## 8. Conclusions

This paper proposed an emotion-based automatic music classification method that can classify music with high accuracy according to commonly identified emotions for use in the development of various types of music-related emotional intelligence IoT services. The well-known circumplex model was used to identify and quantify emotions. Music-correlated emotion identification results for various genres were obtained via a survey of the study participants and music data were extracted by developing the Java-based automation tool. The extracted data was implemented in multiple regression analysis to derive the emotion classification map via SVM. Thus, it was possible to automatically classify the emotions felt when listening to various genres of music.

In our experiments, we compared the emotional analysis results obtained via the proposed method to those obtained via the survey from three different perspectives. The proposed method yielded emotion match rates of 44.71%, 77.06% and 94.12%, whereas the survey yielded match rates of 44.71%, 87.79% and 100%, respectively. This presented that the judgment of emotion by the proposed method was similar to the average judgment of emotion by the survey participants. Thus, this indicates that the proposed method can be effectively utilized as an automatic method to identify and classify the common emotions felt by people when they listen to music. Finally, in our experiment, we

conducted comparative analysis with different classification algorithms to analyze the performance for the emotion match rate.

Although the results from this study are very encouraging, further research will be carried out to address several issues. First, four emotional categories were defined in this study: 'Happy, Glad,' 'Excited, Aroused,' 'Sad, Angry' and 'Calm, Bored'; however, we will further decompose these emotions in order to classify them into more detailed categories. Second, we will apply the proposed method to more songs to investigate the generalization of the results. Finally, we aim to carry out studies that utilize lyrics apart from the wavelength of music, as emotional analysis can vary according to the sound quality of music.

**Author Contributions:** Conceptualization, Y.-S.S.; Data curation, Y.-S.S.; Formal analysis, Y.-S.S.; Funding acquisition, Y.-S.S.; Investigation, Y.-S.S. and J.-H.H.; Methodology, J.-H.H.; Project administration, Y.-S.S.; Resources, Y.-S.S. and J.-H.H.; Software, Y.-S.S. and J.-H.H.; Supervision, J.-H.H.; Validation, Y.-S.S.; Visualization, J.-H.H.; Writing—original draft, Y.-S.S. and J.-H.H.; Writing—review & editing, Y.-S.S. and J.-H.H..

**Funding:** This work was supported by the National Research Foundation of Korea (NRF) grant funded by the Korea government (MSIT) (No. NRF-2017R1C1B5018295). Also, this work was supported by the 2016 Yeungnam University Research Grant.

**Acknowledgments:** This paper is the result of invaluable contributions from many individuals. The authors would especially like to thank Jung-Wook Park and Dong-Hyun Lee for their work on the initial design of the experiment. In addition, we really appreciate each of participants who dedicated time to completing our surveys.

**Conflicts of Interest:** The authors declare no conflict of interest.

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
