# Peer review of "Automatic Emotion-Based Music Classification for Supporting Intelligent IoT Applications"

_electronics, doi:10.3390/electronics8020164_

Round 1

Reviewer 1 Report

- first, music is emotioanlly perceived in differnet ways;

    - a person addicted to classical music, may don't like pop music at all

    - a person addicted to heavy metal, wull never like pop music
the authors should clarify this fact.

- moreover, the study reported is limited. Increase the number participating persons to an statistical significant amount.

- configuration and parameter sets are not given

- compare the support vectore machine-approach to random forrest and deep neural network approach  

Author Response

We sincerely thank the reviewers for their careful reading of this article and for their kind comments, which have helped to improve this manuscript. We revised carefully our manuscript based on your comments. The detailed responses to your comments are as follows:

Comment 1)

1)

- first, music is emotionally perceived in different ways;

    - a person addicted to classical music, may don't like pop music at all

    - a person addicted to heavy metal, wull never like pop music

the authors should clarify this fact.

Answer 1)

We fully agree with your comment. Following   your comment, we have added the contents you pointed out in the ‘6.1.   Experimental Design’ section.

Thank you for your valuable comment.

Revision)

We have added the following explanations to   clarify this point.

(Page 9)

Please note that there are a wide variety of   musical genres and even the same music could be perceived depending on the   person. Although the experiments conducted in this study have focused on   Korean pop songs (K-pops) and as it is possible that those who are addicted   to classical or heavy metal music genre would not enjoy them at all, they   would feel different emotions. The main reason for choosing K-pops is that   they are not only receiving global attention and becoming more popular [42]   but also embrace various musical genres. K-pops were used for the study as   the music in each K-pop genre allows individual listeners to feel a quite   specific emotion and considered to be appropriate for the evaluation of   emotion identification performances.

* References

42. BBC Minute: On how   K-Pop took over the world, Available online: https://goo.gl/wDydkz (accessed   on 18 Jan 2019).

Comment 2)

2)

- moreover, the study reported is limited. Increase the number participating persons to an statistical significant amount.

Answer 2)

As mentioned in the ‘5.1. Experimental Design’   section, we had recruited seven participants for our experiment.

We also think that it is important to increase   the number of participants. Following your comment, we really really tried to   recruit more participants and to increase the number of participants. As a   result, we collected from the emotional data from forty participants.

As the number was increased, we did the entire   experiments again and described the changed intermediate and experimental   results in detail.

Thank you very much for your constructive   comment.

Revision)

We have revised our paper using the changed experimental   results in Section 6.

(Page 8)

In order to investigate the emotions that people feel when   listening to music, we recruited 40 participants hoping to join this   experiment; 31 participants in their 20s, 6 participants in their 30s, and 3   participants in their 40s. In this study, the music   files used to collect data for the survey were 100 Korean pop songs that   gained global popularity. Various genres such as dance, ballad, R&B,   Hip-Hop, and rock were selected. The audio source was an MP3 file with a   sound quality of 320 kbps.

(Pages 10-11)

Figure 8. Data distribution for   valence with each independent variable.

Figure 9. Data distribution for   arousal with each independent variable.

Valence = (     -209.7)+ ( 0.004548 * Average height)+ ( 0.0005603 * Peak average)+ ( 2.29 * BPM)(1)

Arousal = (-76.228036)+ (0.012440 * Average height)+ (-0.006034 * Peak average)+ (0.240159 * HfW)+ (-0.005287 * Average width)(2)

(page 12)

Figure 10. Emotion Classification   Map.

(pages 12-13)

Table 3 shows the probability (%) of occurrence of the same   emotion by comparing the results from the proposed method to the results from   the survey participants, in each 20-s interval section. In Table 3, TPM (The   Proposed Method) refers to the method proposed in this study, and A to AN are   the survey participants. The numbers in each column show the emotion match   rate between the results obtained via TPM and those obtained via the survey   participants. AVG is the average value of the emotion match rate for TPM and   each participant. The bold numbers indicate the best emotion match rate   between TPM and each participant. All values are rounded off to 3 decimal   places.

Table 3. Emotion match rate.

(%)Comparison of TPM and survey participants (A to AN, 40 participants)TPMA72.94B74.71C74.12D74.71E75.88F74.12G72.94H72.94I71.76J72.94K75.29L72.94M74.71N74.12O73.53P72.35Q75.88R77.65S72.35T72.35U72.94V71.76W71.18X75.88Y76.47Z77.65AA72.35AB72.94AC75.29AD74.71AE74.71AF70.59AG74.12AH72.94AI73.53AJ74.12AK75.88AL75.29AM75.29AN72.35AVG: 73.96

As is shown in Table 3, the average match rate was 73.96%   and the maximum match rate was 77.65% obtained from R and Z. The match rates   obtained via surveying participants were found to range from 70.59% to   77.65%.

(page 13-14)

Table 4. Emotion match rates for   the “All”, “Most”, and “Least” cases.

(%)TPMABCDEFAll44.7144.7144.7144.7144.7144.7144.71Most77.0687.0687.0687.6590.0082.3587.65Least94.12100.00100.00100.00100.00100.00100.00 GHIJKLMAll44.7144.7144.7144.7144.7144.7144.71Most85.8887.0688.2492.3588.8286.4790.00Least100.00100.00100.00100.00100.00100.00100.00 NOPQRSTAll44.7144.7144.7144.7144.7144.7144.71Most85.8885.8890.0090.0089.4188.8286.47Least100.00100.00100.00100.00100.00100.00100.00 UVWXYZAAAll44.7144.7144.7144.7144.7144.7144.71Most88.8290.0085.8888.8286.4785.2984.71Least100.00100.00100.00100.00100.00100.00100.00 ABACADAEAFAGAHAll44.7144.7144.7144.7144.7144.7144.71Most87.6590.0088.8289.4189.4188.2487.06Least100.00100.00100.00100.00100.00100.00100.00 AIAJAKALAMAN-All44.7144.7144.7144.7144.7144.71-Most83.5388.8287.6588.2485.2990.59-Least100.00100.00100.00100.00100.00100.00-

As presented in Table 4, in the case of “All”,   the emotional match rate was the same as 44.71% for TPM and each participant.   Because it was an uncommon occurrence for all of the survey participants to   always feel the same emotion each other, the match rate was not high. Note   that, in each song, there was no case where only the emotional result derived   via TPM was different when the emotional results derived via survey   participants were all the same as one emotional type. That is, the low match   rate was caused by one or more different emotional results via survey   participants. Because emotion perception is essentially subjective, one could   perceive different emotions even when listening to the same song. Thus, the   following “Most” could be more valid experimentation strategy. In the case of   “Most”, the match rate via TPM was 77.06%. That is, TPM could identify fairly   large of the emotions felt by the participants (the emotion classification   results obtained via TPM are in agreement at a rate of 77.06% with the most   commonly classified emotions among the results of the survey). In the case of   “Least”, the match rate via TPM was 94.12%. That is, TPM could identify at   least one of the emotions that were felt by many participants (the emotion   classification results obtained via TPM are in agreement at a rate of 94.12%   with one of the emotions among the results of the survey).

(pages 14-15)

Figure 12. Comparison of the   emotion match rates.

For the three cases, as presented in Figure 12, the   emotion match rates obtained via TPM were 44.71%, 77.06%, and 94.12%, whereas   the average emotion match rates from the survey were 44.71%, 87.79%, and   100%, respectively. These results demonstrate that the emotions via the   survey participants are largely identified via TPM. Of these cases, the match   rate via TPM from the perspective of “Most” was 77.06%, which is   approximately 10% lower than the 87.79% average obtained via the survey   participants. Although the rate of TPM is a little lower than that of the   survey participants, this confirms that TPM can be effectively utilized as an   automatic method to identify and classify the common emotions felt by people   when they listen to songs.

Comment 3)

3)

- configuration and parameter sets are not given.

Answer 3)

Following your comment, we have described the   configuration and parameter sets in Section 6.1.

Thank you for your valuable comment.

Revision)

We have revised our paper by describing specifically the parameters   in Section 6.1.

(Page 11)

The   parameter setting is as follows: Kernel is linear, Tolerance is 0.001   (tolerance of termination criterion), and epsilon is 0.1 (epsilon in the   insensitive-loss function).

(page 15)

The   configuration and parameter sets are as follows:

·             SVM: Kernel   is linear, Tolerance is 0.001 (tolerance of termination criterion), and   epsilon is 0.1 (epsilon in the insensitive-loss function)

·             Random   forest: the coefficient(s) of regularization is 0.8 and regularization is   true (1)

·             Deep neural   network: hidden dim is 30 (the dimension of each layer), learning rate is   0.1, epoch num is 1,000 (1,000 iteration over the entire dataset), and hidden   layer is 3

·             K-nearest   neighbor: K is set from 2 to 6

 Comment 4)

4)

- compare the support vectore machine-approach to random forrest and deep neural network approach.

Answer 4)

We also feel that it is important to compare   the support vector machine approach with the different classification algorithms,   such as, random forest and deep neural network. Thus, with your comment 2, we   have conducted the comparative analysis using the experimental results with   the increased number of participants. In the analysis, we have also applied   K-nearest neighbor that is one of the general classification algorithms.

Thank you for your constructive comment.

Thank you very much for your incisive and constructive comments. They will be very helpful towards making progress with our further work. Thank you again.

Reviewer 2 Report

Automatic Emotion-based Music Classification for Supporting Intelligent IoT Applications

------------------

A survey for collecting emotional data is conducted based on the emotional model. In addition, music features are derived by discussing with the working group in a small and medium-sized enterprise. Emotion classification is carried out using multiple regression analysis and support vector machine.

Major revision:

Desired modifications

a) Numerical comparison with other related studies.

b) Description on an IoT application

If b) is not aplicable, please omit the IoT in the title and abstract.

If a) is not possible, please justify why?

---------------------

RESEARCH ISSUES

---> In the paper you mention:

"Generally, based on the various acoustic features of music, many studies used different classifier to recognize music emotion, such as K-nearest neighbors, support vector regression, and support vector machines. Unlike the previous studies, we identified new music features by discussing with the working group..."

why not a comparison with such studies? or justify why they are not aplicable?

---> the paper is partially focused on IoT application.... why not mention such application? or benefits of your study with regard to IoT applications.

Only you mention IoT in Introduction, Related work and conclussions. 

It would be great to have a section about an IoT application.

---> Figures 1, 2 and 3. add more information in captions, including reference to Russell's work, etc...

---> Figure 5. more information in caption

LAYOUT ISSUES

---> Line 69.- remove subsection 2.1 caption

---> Joint figures 13 and 14 in one, using the whole textwidh.

ENGLISH EDITING

---> Use impersonal form ... as am example:

line 126 "...we identified..." replace by "...new music features were identified ..."

---> some minor typos

Author Response

We sincerely thank the reviewers for their careful reading of this article and for their kind comments, which have helped to improve this manuscript. We revised carefully our manuscript based on your comments. The detailed responses to your comments are as follows:

========================================================

A survey for collecting emotional data is conducted based on the emotional model. In addition, music features are derived by discussing with the working group in a small and medium-sized enterprise. Emotion classification is carried out using multiple regression analysis and support vector machine.

Major revision:

Desired modifications

a) Numerical comparison with other related studies.

b) Description on an IoT application

If b) is not applicable, please omit the IoT in the title and abstract.

If a) is not possible, please justify why?

Comment 1)

1)

---> In the paper you mention:

"Generally, based on the various acoustic features of music, many studies used different classifier to recognize music emotion, such as K-nearest neighbors, support vector regression, and support vector machines. Unlike the previous studies, we identified new music features by discussing with the working group..."

why not a comparison with such studies? or justify why they are not applicable?

Answer 1)

We also think that it is important to compare   the support vector machine approach in our study with the different   classification algorithms. Thus, in this revision, we have applied the   popular classification algorithms in our experiment, such as K-nearest   neighbors, random forest, and deep neural network.

Moreover, we have really tried to increase the   number of participants from seven to forty, to improve the reliability of the   experimental results.

Finally, we have conducted the comparative   analysis using the experimental results with the increased number of   participants.

Thank you very much for your constructive comment.

Revision)

In Section 6.3., we have added the experimental   results for random forest, deep neural network, and K-nearest neighbor, based   on the survey data collected from forty participants.

(Page 15)

In the   proposed method, SVM is used to identify the emotional categories and   classify the emotions for the music. However, as is well known, there are   several different classification algorithms aside from SVM. Thus, in our   experiment, we additionally analyze the proposed method with popular   classification algorithms, such as random forest [45], deep neural network   [46], and K-nearest neighbor [33, 46], as well as SVM. The configuration and   parameter sets are as follows:

·             SVM: Kernel   is linear, Tolerance is 0.001 (tolerance of termination criterion), and   epsilon is 0.1 (epsilon in the insensitive-loss function)

·             Random   forest: the coefficient(s) of regularization is 0.8 and regularization is   true (1)

·             Deep neural   network: hidden dim is 30 (the dimension of each layer), learning rate is   0.1, epoch num is 1,000 (1,000 iteration over the entire dataset), and hidden   layer is 3

·             K-nearest   neighbor: K is set from 2 to 6

The   overall experimental results are shown in Table 5. In Table 5, the first   column indicates the survey participants, and the other columns indicate the   experimental results from classification algorithms. RF means random forest,   DNN means deep neural network, and KNN denotes K-nearest neighbor. The bold   and underlined numbers indicate the best emotion match rate in each   participant. All values are rounded off to 3 decimal places.

As presented   in Table 5, for all of the survey participants, 32 out of 40 participants   showed the highest emotion match rate when using SVM. Next, 11 out of 40 participants   presented the best emotion match rate when using DNN. RF and KNN (K=5) showed   the best rate for the 1 participant (E and P).

Figure 13 provides   the average match rate of the proposed method with the various classification   algorithms. The best average match rate was 73.96% obtained via SVM, and the second   one was 72.90% obtained via DNN. The lowest average match rate was 64.12% via   KNN (K=2).

Table 6 shows   the emotion match rate according to "All", "Most", "Least”   with the classification algorithms. The bold and underlined numbers indicate   the best emotion match rate for each perspective. As shown in Table 6, compared   with the other classification algorithms, SVM showed a better performance for   each of the three perspectives.

Table 5. Comparative analysis   for emotion match rate with classification algorithms.

Figure 13. Average match rate from   classification algorithms.

Table 6. Emotion match rate for   the “All”, “Most”, and “Least” cases with classification algorithms.

(%)SVMRFDNNKNNK=2K=3K=4K=5K=6All44.7141.18 43.53 38.24 41.18 41.18 42.35 40.00 Most77.0672.35 74.71 66.47 70.59 71.76 72.35 70.00 Least94.1290.59 94.12 86.47 89.41 90.00 90.59 88.82

Comment 2)

2)

---> the paper is partially focused on IoT application.... why not mention such application? or benefits of your study with regard to IoT applications.

Only you mention IoT in Introduction, Related work and conclusions.

It would be great to have a section about an IoT application.

Answer 2)

We fully agree with your comment. Following   your comment, we have created a new section about an IoT application.

Thank you for your valuable comment.

Revision)

We have added the following section for this   issue.

(Page 2)

2. Intelligent   IoT Applications

Following the stream of the 4th Industrial   Revolution, the IoT technology is becoming more and more intelligent and   transforming into a high advanced stage [15]. By communicating and   collaborating with each other, the IoT equipment pervades our daily lives   through IoT-related services. Also by becoming more sophisticated, they are   gradually realizing the future pictures we have been envisioning. As such,   people’s expectations and interest are rising as much as the value IoT   technology is expected to create in our lives and industries.

The products embedded in IoT technology are   being utilized in many ways and categories [16]. Especially, they are   providing much convenience in our everyday life by becoming more intelligent   over time. Global companies are investing huge research funds to develop   advanced intelligent IoT applications for the areas involving healthcare,   smart car, smart home, and smart factory systems. For instance, they are   making greater efforts to provide a more intelligent service for smart   devices (e.g., smart TV, smart phone, smart pad, smartwatch, smart glass,   smart speaker, etc.) which are easily accessed or widely used by general   users.

In particular, since the intelligent personal   assistant (IPA) [17] such as Amazon Alexa [4-5], Apple Siri [6-8], Google   Assistant [9-10], or Microsoft Cortana [11-12] has been introduced, we are   receiving more intelligent services. Currently, these IPA’s aim to provide a   wide range of services by consistently collecting various kinds of   information including time, location, and personal profiles. They provide a   typical web search result when they are unable to reply to the user’s   question clearly or directly. The current IPA’s basically provide information   to the users in two ways, proactively or reactively [17-18]. The proactive   assistant refers to the agent who automatically provides information useful   to the user without his/her explicit request. That is, it recommends a   specific service depending on the upcoming event or the user’s current   situation. The latter refers to an agent who directly responds to the   question or the sentence actually entered by the user.

User’s information is consistently collected,   accumulated, and utilized more following the development of intelligent   technologies and now, aside from collecting general information, the   technologies which can provide appropriate services by identifying or   recognizing user’s emotion and achieve adequate interactions are being   studied for commercialization. In this regard, one of the IoT-related   technology fields which can easily be understood or accessed by the users is   the IoT technology utilizing music. For example, certain music can be   provided to the user through a smart device to satisfy his/her desire after   understanding his/her emotional state, or today’s weather conditions.

Thus, this study attempts to focus on the   R&D of a platform technology which will be able to automatically   categorize the emotions associated with various types of music to support an   intelligent IoT application technology which satisfies the smart device   user’s emotions or emotional needs and then recommend/provide proper music or   musical genre accordingly.

* References

15. Abramovici,   M.; Gobel, J.C.; Neges, M. Smart Engineering as Enabler for the 4th   Industrial Revolution; Integrated Systems: Innovations and Applications;   Springer: Cham, Switzerland, 2015; pp. 163-170.

16. Vidal-Garcia,   J.; Vidal, M.; Barros, R. H. Computational Business Intelligence, Big Data,   and Their Role in Business Decisions in the Age of the Internet of Things;   IGI Global: Hershey, PA, USA, 2019; pp. 1048-1067.

17. Sarikaya, R.   The Technology Behind Personal Digital Assistants: An overview of the system   architecture and key components. IEEE Signal Processing Magazine 2017, 34,   67-81, 10.1109/MSP.2016.2617341.

18. Yorke-Smith,   N.; Saadati, S.; Myers, K. L.; Morley, D. N. The design of a proactive   personal agent for task management. International Journal on Artificial   Intelligence Tools 2012, 21, 1250004, 10.1142/S0218213012500042.

Comment 3)

3)

---> Figures 1, 2 and 3. add more information in captions, including reference to Russell's work, etc...

Answer 3)

Following your comment, we have added more   descriptions in captions for Figure 1, 2, and 3.

Thank you for your valuable comment.

Revision)

We have revised the captions for Figure 1, 2,   and 3.

(Page 3)

Figure 1. Russell’s circumplex model. The circumplex model is   developed by James Russell [19]. In the model, emotions are distributed in a   two-dimensional plane. The x-axis represents valence and the y-axis   represents arousal. Valence refers to the positive and negative degree of   emotion, and arousal refers to the intensity of emotion. Using the circumplex   model, emotional states can be presented at any level of valence and arousal.

(Page 4)

Figure 2. Thayer's model. The Thayer’s model is proposed by Robert   E. Thayer [22]. In the model, emotional states are represented by energy and   stress corresponding to arousal and valence of the circumplex model. Energy   refers to the volume or intensity of sound in music, and Stress refers to the   tonality and tempo of music. The mood is divided into four clusters:   calm-energy (e.g., Exuberance), calm-tiredness (e.g., Contentment),   tense-energy (e.g., Frantic), and tense-tiredness (e.g., depression). Using   the Thayer's model, it is possible to characterize musical passages according   to the dimensions of energy and stress [25].

(Page 5)

Figure 3. Tellegen-Watson-Clark model. In the model, high positive   affect and high negative affect are proposed as two independent dimensions,   whereas pleasantness–unpleasantness and engagement-disengagement represent   the endpoints of one bipolar dimension [26].

* References

25. Music Cognition Handbook: A   Dictionary of Concepts. Available online: https://goo.gl/5vYCG4 (accessed on   18 Jan 2019).

26. Watson, D.; Tellegen, A. Toward a   consensual structure of mood. Psychological Bulletin 1985, 98, 219-235,   10.1037/0033-2909.98.2.219.

Comment 4)

4)

---> Figure 5. more information in caption

Answer 4)

Following your comment, we have added more   descriptions in caption for Figure 5.

Thank you for your valuable comment.

Revision)

We have added more information in the caption of Figure 5.

(Page 6)

Figure 5.   Overall approach. The approach consists of three steps: (1) emotional and   music data are collected from participants and songs, (2) regression analysis   is applied to identify the relationship between emotional data and music   data, and (3) a classification algorithm is applied to determine how the   emotional data correlated with the music data are related to the actual   emotions.

Comment 5)

5)

---> Line 69.- remove subsection 2.1 caption

Answer 5)

We carefully think that subsection 2.1 caption could   be left because there is subsection 2.2 (3.2 in the revised manuscript). If   you still think that subsection 2.1 should be removed, we will do that.

Thank you for your valuable comment.

Comment 6)

6)

---> Joint figures 13 and 14 in one, using the whole textwidh.

Answer 6)

We have arranged the figure 13 and 14 (figure   14 and 15 in the revised manuscript) in the whole textwidth.

Thank you for your valuable comment.

Revision)

(Page 17)

Figure 14. Waveform of the high-quality version of S.Figure 15. Waveform of the low-quality version of S.

Comment 7)

7)

---> Use impersonal form ... as am example:

line 126 "...we identified..." replace by "...new music features were identified ..."

Answer 7)

We have changed the sentences into the   impersonal form.

Thank you for your valuable comment.

Revision)

We have revised following your comment.

(Page 5)

Unlike the previous studies, in our study, new music   features were identified by discussing with the working group for developing   intelligent IoT applications in a small and medium-sized enterprise. Also, a   combined approach with regression and support vector machine was applied.   Moreover, through the survey, induced (evoked) emotion was used for the   validation of the proposed method.

Comment 8)

8)

---> some minor typos

Answer 8)

During the revision of this manuscript, we have   corrected minor typos in the entire paper.

Thank you for your valuable comment.

Thank you very much for your incisive and constructive comments. They will be very helpful towards making progress with our further work. Thank you again.

Round 2

Reviewer 2 Report

All proposed changes have been made.

References are updated.

Figures and tables captions are improved.

....